# A Profile of Knee Injuries Suffered by Australian Army Reserve Soldiers

**DOI:** 10.3390/ijerph16010012

**Published:** 2018-12-20

**Authors:** Ben Schram, Robin Orr, Rodney Pope

**Affiliations:** 1Tactical Research Unit, Bond University, Robina 4226, QLD, Australia; rorr@bond.edu.au (R.O.); rpope@csu.edu.au (R.P.); 2School of Community Health, Charles Sturt University, Albury 2640, NSW, Australia

**Keywords:** reserves, part-time, military, health and safety, defence, injury

## Abstract

Despite having to perform the same occupational tasks as full-time soldiers, part-time soldiers may have lower levels of physical conditioning and report higher rates of injury per unit exposure to active service. The purpose of this study was to profile the leading body site of injury occurring in part-time soldiers to inform injury prevention strategies. Injury data from the Australian Army Reserve (ARES) spanning a two-year period were obtained from the Department of Defence Workplace Health, Safety, Compensation and Reporting database pertaining to locations, nature, mechanisms, and the activity being performed at the time of injury. Among the 1434 injuries reported by ARES personnel, the knee was the most common injury site (*n* = 228, 16%). Soft tissue injury due to trauma or unknown causes was the most common nature of knee injury (*n* = 177, 78%). Combat training was the most common activity being performed when soft tissue injuries occurred at the knee (*n* = 73, 42%), with physical training the second most common (*n* = 51, 30%), due to muscular stress (*n* = 36, 71%) and falls (*n* = 8, 16%). Targeted intrinsic and extrinsic approaches to injury minimization strategies for soft tissue knee injuries during combat and physical training should be designed.

## 1. Introduction

An effective military force is required to be agile, capable, efficient, and potent. Injuries to military personnel interrupt active duty service and detract from overall army capability [1]. Emphasis is on ensuring injury prevention strategies are put in place to minimise injuries sustained during military service [2,3,4]. To be effective however, these intervention strategies need to be informed by research investigating key types and sources of injuries, so that risk mitigation strategies are directed towards the most prevalent or serious types and sources of injury [2,3,4].

The Australian Defence Force is comprised of both part-time and full-time personnel [5]. Even though part-time personnel are not employed on a full time basis, these members are important contributors to the Army’s capabilities [6]. With part-time members having similar training requirements, fitness standards, and occupational tasks as full-time personnel; injuries pose similar concerns for this population. Injuries may be of even greater concern for part-time soldiers, as previous research has shown part-time personnel report a substantially higher rate of injuries per unit exposure to active duty when compared to their full-time counterparts, with some showing up to double the rates in part-time compared to full-time personnel [7,8].

One possible reason for a heighted risk of injury in part-time personnel may be lower levels of aerobic fitness in part-time personnel [9,10,11]. Part-time personnel may have a lower fitness level when compared to full time personnel, possibly due to the necessity to balance both civilian life and military life [9]. It is well known that lower aerobic fitness is a risk factor for injuries in military personnel [10,12].

The lower limb has been shown to be the most commonly injured body site in military personnel [13]. However, further detail regarding the particular sites in the lower limb that are most frequently injured and the activities and mechanisms that give rise to injuries affecting those sites is lacking for part-time personnel [13]. This detail is needed to inform injury prevention efforts for part-time personnel and so enhance the readiness of this important element of Army capability. A detailed investigation of the nature and cause of injuries affecting specific lower limb sites will valuably inform selection and implementation of interventions designed to minimise both the incidence and severity of injuries suffered by the part-time army population. This approach is supported by previous research in the army context, which has provided positive evidence for injury rate reduction through targeted interventions, once high risk activities and locations have been identified [2,14,15,16]. The aim of this study was to profile the most common injury occurring in part-time soldiers with respect to the predominate location, nature, activity, and mechanism, with a view to informing injury prevention strategies for these part-time personnel.

## 2. Materials and Methods

The study employed a cross-sectional design, using Australian Department of Defence population data and records of injuries reported by Australian Army Reserve (ARES) personnel over a two-year period. The mean annual population size for the ARES across the period (*n* = 15,034) was ascertained from published Department of Defence Records [17,18]. The injury records for ARES personnel were sourced in a non-identifiable form from the Workplace Health, Safety, Compensation and Reporting (WHSCAR) database of the Australian Department of Defence, which constitutes the official record of incidents and injuries sustained by Army personnel [19]. The WHSCAR records provided to the researchers included, for each injury to an ARES soldier: details of the Service (Army) and service type (part-time), the type of occurrence the record reported (serious injury or minor injury), the injury date, the body site, the nature of the injury, the activity being undertaken at the time of the injury, the mechanism of injury, the soldier’s duty status at the time of injury (on duty or off duty), age and rank, and a narrative description of the injury incident.

The injury records extracted from the WHSCAR database were included in the study if they related to: (1) ARES personnel; and (2) a minor or serious injury that occurred while the soldier was ‘on duty’. The definitions of injury, whether for minor personal injury (MPI) or serious personal injury (SPI), were those provided by the Australian Department of Defence [19]. An SPI required immediate treatment as an in-patient in a hospital, whereas an MPI was defined as any minor injury which did not result in a fatality, SPI, illness, or dangerous incident. Records were excluded if they: (1) related to personnel from military services other than the Australian Army; (2) related to full-time Army personnel; (3) related to personnel from a foreign defence force, on secondment with the Australian Defence Force; or (4) contained missing or incomplete data pertinent to this study (for example, site of injury).

The WHSCAR data were manually cleaned following receipt to ensure that only eligible records were retained. Each record was reviewed, and duplicate and ineligible records were removed; the latter with reference to the inclusion and exclusion criteria listed above. Each datum record was further verified, corrected, or made more precise by manually comparing the allocated Type of Occurrence Classification System (TOOCS) classifications with the free text narrative data from the same record. When discrepancies were identified, precedence was given to the free text narratives and the TOOCS classification was adjusted accordingly, as narratives provided by incident reporters are considered more detailed and accurate than data entered by a third party using a finite coding system [20]. The adjusted dataset was employed in the data analysis. To increase data accuracy, brevity, and sensitivity, some TOOCS fields were aggregated, notably the ‘nature of injury’ and ‘activity’ fields. In the TOOCS ‘nature of injury’ classifications, ‘soft tissue injuries due to trauma or unknown mechanism’ subsumed ‘trauma to muscle’ (a soft tissue) and ‘trauma to tendons’ (another soft tissue). In addition, ‘trauma to joints and ligaments’ subsumed ‘trauma to joints and ligaments, not elsewhere classified’ and ‘trauma to joints and ligaments unspecified’. In the TOOCS ‘activity’ classifications, all sports were merged into an aggregated classification, ‘sport’. ‘Running’ was subsumed by ‘physical training’, and ‘patrolling’ and all weapon handling activities were subsumed by ‘combat training’. 

Data Analysis

Following this data cleaning process, descriptive analyses were employed to examine and describe the data to address the study aim. Injury incidence rates were first calculated as follows: the counts of minor injuries and serious injuries reported by ARES personnel across the two-year study period were each divided by the number of full-time-equivalent years of active service provided by ARES personnel across these two years and the result was then multiplied by 100 to derive the number of injuries reported by ARES personnel per 100 years of active service. In these calculations, one full-time-equivalent year of active service by ARES personnel was defined as 232 days of active service, based on the following:

Total days of active service in one full-time year = 365 days in a full year − 104 days of weekends (or equivalent days ‘in lieu’) − 20 days of annual leave − 9 days of public holidays = 232 days.

The proportion of reported occupational injuries in ARES personnel which occurred at each of the reported body locations was next calculated and these proportions were tabulated to enable identification of the most common body site of injuries in ARES personnel. Subsequently calculated were the proportions of knee injuries reported by ARES personnel which were associated with personnel in specific age brackets (less than 20 years, 20–29 years, 30–39 years, 40–50 years and greater than 50 years) or at specific rank levels, or which involved specific: (1) natures of injury; (2) activities being performed at the times injuries occurred; and (3) injury mechanisms. The latter proportions were used to identify the key natures, causes, and mechanisms of knee injuries reported by ARES personnel.

The Australian Defence Human Research Ethics Committee (ADHREC, LERP14-024) and the Bond University Human Research Ethics Committee (BUHREC, RO-1907) granted ethics approval for this study. As this data were retrospective and non-identifiable, it met the pre-requisites for a waiver of participant consent as there were no means through which the participants could be identified, and consent gained. Departmental authorisation for the project was obtained in parallel to the process for obtaining ADHREC approval. Authorization to publish this study was obtained from Joint Health Command.

## 3. Results

A total of 1434 injuries were reported by the ARES population of 15,034 personnel within the 2-year study period, representing a reported injury incidence rate of 30.5 injuries per 100 person-years of active service. Minor personal injuries (MPIs) accounted for 95.4% of these reported injuries (*n* = 1368), with the remaining 4.6% (*n* = 66) being classified as serious personal injuries (SPIs). The body sites of all lower limb MPI and SPI reported by ARES personnel in the two-year study period are listed in Table 1.

The knee was the most commonly reported specific body site of injury, accounting for 15.9% (*n* = 228) of all injuries reported by ARES personnel (Table 1) and therefore will be the focus for the rest of this investigation. Of the total 228 knee injuries reported by ARES personnel, 98% (*n* = 224) were MPIs and 2% (*n* = 4) were SPIs (Table 1). The SPIs which occurred at the knee were due to dislocation, one during parade and one while jogging.

The nature of soft tissue injuries suffered by part-time personnel can be found in Table 2 below. The most common nature of knee injuries reported by the ARES personnel was soft tissue injury due to trauma or unknown mechanisms, which was responsible for 81% of the injuries in this bodily location. This was followed by trauma to the joint and ligaments.

The activity being performed while soft tissue injuries were suffered at the knee can be found in Table 3 below. Combat training was the leading activity in which injuries occurred at the knee with 43% of injuries, while physical training was the second most common. Combining both combat training and physical training led to a total of 72% of soft tissue knee injuries and will therefore be explored further. 

The most common mechanisms of injury reported by ARES personnel for soft tissue injuries at the knee joint that occurred while they undertook combat training were falls and muscular stress with no object being handled (*n* = 27, 37.0% and *n* = 21, 28.8%, respectively; Table 4). Similarly, ARES personnel reported muscular stress with no objects being handled and falls as the most prevalent mechanisms causing soft tissue injuries at the knee joint during physical training, accounting for 70.6% (*n* = 36) of the soft tissue injuries reported during physical training (Table 5).

### Profiles of the Part-Time Soldiers Who Reported Knee Injuries

The mean age of the ARES personnel who reported knee injuries was 33.7 ± 11.1 years, ranging from 18 to 63 years of age, and the age and military rank distributions for the ARES personnel who reported these knee injuries are provided in Table 6 and Table 7. The results presented in Table 6 indicate that while 43% of knee injuries reported by ARES personnel were reported by personnel in the 20–29-year age bracket, 54% of the reported knee injuries were reported by ARES personnel over the age of 30 years. Non-officers reported 183 of the knee injuries reported by ARES personnel (80.3%) while officers (including officer cadets) reported 45 of the knee injuries (19.7%; Table 7). Also, of note, 25% of the knee injuries reported by ARES personnel were reported by recruits, trainees and officer cadets and would therefore appear to have occurred during ARES initial training courses (Table 7).

## 4. Discussion

The aim of this study was to identify the leading predominate locations, natures, activities, and mechanism of injuries occurring in part-time army personnel with a view to informing injury prevention strategies. This is the first known study to provide a detailed analysis of the leading body sites, natures, and causes of injuries reported by part-time army personnel. A key finding of this study was that the knee was the most commonly reported body site of injury in ARES personnel, accounting for 16% of all injuries reported by these part-time army personnel. Part-time personnel (30.5 injuries per 100 years of active service) appear to be injured more than their full-time colleagues with previous reports of full-time personnel injury rates of 16.72 injuries per 100 years of active service [7]).

This finding is in agreement with previous studies in military personnel, with injuries occurring at or below the knee most common [2,10,11,21]. Kaufman’s review [2] included seven studies in which knee injuries comprised 10.2–34.3% of all reported injuries in a variety of military recruits including marines, army and naval special warfare candidates. Knapik found the knee to be the location of 21% of injuries for males and 19% for females in an army population, a similar finding to this current study [10]. In Knapik’s study, the ankle and foot (20%) were commonly injured, in line with the findings of this study in which the ankle was the third most common site of injuries, accounting for 9.83% of all reported injuries (Table 1). Jennings found 18% of injuries in army soldiers were to the knee [21], while previous research on Australian military recruits has found that 24% of injuries occurred at the knee [11]. Across the Australian Army as a whole, aggregated injury data showed that knee injuries accounted for the highest proportion of lower limb injuries (35%) associated with a single body site and the highest proportion of working days lost (40%) [8].

Soft tissue injuries were the most common nature of injuries in the part-time, ARES population that was the subject of the current study, and these soft tissue injuries were most commonly due to either trauma or unknown mechanisms. It should be acknowledged however that the categorization of the nature of injury soft tissue injuries (trauma or unknown mechanisms) may be broad in nature and therefore may be over-represented because of this. Soft tissue injuries at the knee have previously been found to be the predominant type of injury amongst active duty military personnel, most commonly involve damage to structures such as the patella, meniscus, and ligaments [22]. The risk factors for soft tissue injuries affecting the knee amongst military personnel may include a history of prior injury to the affected knee, prior deployments, infantry service, and prior hip injury [22]. Hill et al. [22] also identified age greater than 30 years and increasing length of service to be risk factors for knee injuries. While this study similarly found that personnel over the age of 30 years featured heavily among those reporting knee injuries in this part-time, ARES population (53.51% of knee injuries), higher military ranks (and potentially lengths of service) did not appear to be strongly associated with higher injury rates.

In this study of part-time army personnel from the ARES, military combat training and physical training were the most common activities being performed when soft-tissue injuries occurred to the knee. Combat training is inherently more difficult for part-time personnel to prepare for. Unlike physical training which they can perform in their own time, combat training requires access to other personnel, restricted equipment, and preferably military styled tasks [23]. An example would be combat load carriage, which would preferably involve carrying their actual military loads and equipment [24]. This requirement is of importance given that military personnel are required to carry heavy combat loads, which can be in excess of 45 kg [25], and carrying these loads is a known cause of serious knee injuries. In a study of military soldiers conducting a 20-km march with a 46-kg load, Knapik et al. [26] observed an injury frequency of one per cent (two of 335). These two knee injuries equated to 14 days of limited duties. A subsequent study by Knapik et al. [27] found that, of six soldiers who could not complete a series of load carriage trials (six 20-km marches with loads ranging from 34 to 61 kg and utilising two difference pack designs), three were diagnosed with a knee strain, with the remaining three suffering injuries across three other body sites. To maintain specific fitness for combat load carriage, it is recommended that a load carriage training session using combat loads is conducted every seven to 14 days [28,29]. Considering this, it is noteworthy that part-time army personnel like those in the ARES may struggle to maintain this level of specific load carriage training, since having ARES personnel carrying military style equipment through urban areas in order to maintain their load carriage conditioning is likely to generate serious concerns among the general public, given current security threats.

Noting the importance of physical fitness for military personnel, it is of concern that physical training is a common source of injuries in both the general military population [8] and in this study. Injuries in physical training are often associated with sudden increases in training volume or intensity [30]. As such, while part-time personnel may be able to maintain some personal fitness, they may be at greater risk of injuries during unit physical training than full-time personnel due to undertaking lower volumes of military physical training than their full-time colleagues. Opportunities to increase military physical training are therefore needed for part-time personnel, potentially utilizing novel approaches like community-based training groups, where ARES personnel can be motivated to train in small groups, connected by close proximity or virtually, in order to maintain a suitable minimum level of fitness [23].

The most common mechanisms of injury during combat training and physical training were falls and muscular stress with no objects being handled. Injuries due to both microtrauma and macrotrauma appeared to occur during combat training in these part-time army personnel, whereas soft tissue injuries with no external stresses, such as overuse injuries, were more prevalent in physical training. Other studies within a military context have found injuries due to running and falls to be common [21], along with a high prevalence of overuse injuries due to repetitive activities [2], with some authors finding overuse mechanisms to be the cause of 75% of injuries in males recruits and 78% in females recruits [10].

Knee injuries in sport are known to occur more frequently in the latter stages of competition under conditions of fatigue, thought to be due to decreased neuromuscular control. Proprioceptive training for the lower limb with embedded cognitive challenges may therefore have a role in the minimization of ankle and knee injuries in those who have been identified with poor proprioception or a previous injury. Programs consisting of wobble boards and jump landing training have shown success in improving both ankle and knee proprioception [31], with wobble boards showing an optimization of ankle joint stability in landing tasks [32]. Poor landing mechanics have been linked to acute ankle and knee sprains [33] Generally increasing levels of fitness and endurance will enable soldiers to operate under less fatigue at the same relative work load, which may also play a role in injury reduction. 

Investigating prevention strategies for overuse injuries in military personnel, a meta-analysis by Kollock et al. [34] has found a possible link between lower hip and thigh strength and overuse injuries of the knee. Despite focusing mainly on patello-femoral pain, the study found decreased hip external rotator, knee extensor, and knee flexor strength in symptomatic personnel, when compared to asymptomatic personnel. While acknowledging that physical training itself may be a cause of injury, it is possible that dedicated and specific strength and conditioning of the hip and knee of military personnel may be useful in minimizing these strength deficits in military personnel, as demonstrated in sporting populations [35]. For the lower limb in general, other injury prevention measures that may be useful in reducing lower limb loading during physical training include altering running surfaces [36], changing training loads [4], bracing ankles for certain activities [3], and use of shock-absorbing innersoles [2].

Not all injury prevention strategies are effective however. There is some evidence suggesting that pre-exercise static stretching, may be ineffective at reducing injury rates in military populations [11]. Likewise, the addition of balance and agility programs on top of physical training has been shown to increase injury rates when implemented in military recruits [37]. It is thought this may be due to this training being offered in addition to normal physical training within an already compact training schedule, leading to extra fatigue and a subsequent increase in injury risk [37]. This study adds to the current body of literature regarding ARES personnel and highlights the top site in which injuries occur among this service type. Differences in injuries between service type may be explained by less chronic conditioning, fewer opportunities to expose personnel to combat training, and time constraints to training in ARES personnel when compared to ARA personnel. Targeted approaches to injury prevention, such as specific physical training, load reduction, avoiding repetitive tasks, and use of ankle bracing may be of use in preventing these injuries. The findings of this study can be used to inform injury prevention efforts and future research for part-time army populations such as the ARES. 

A limitation of this study was that data were extracted from a system in which injury information was collected by use of a formal injury report. This may have led to minor, or less severe injuries not being reported and therefore captured in this reporting system. A recent study has suggested that only 11–19% of injuries are captured with the WHSCAR system [7]. Under this system, the casualty is required to report to a system retrospectively and therefore the numbers of injuries reported in this study may underrepresent the absolute number of injuries in this population. It does remain likely that the proportional distributions of body sites, natures, and causes of injuries reported by ARES personnel are still representative of this part-time army population. There have been recommendations for a ‘point of care’ system of reporting to be utilized, in which health care personnel create a report at the time an injured soldier presents for health care, rather than relying on the casualty to report to a system retrospectively [7].

## 5. Conclusions

Soft tissue injuries affecting the knee were found to be the most common nature and site of injury amongst the part-time, ARES personnel. These injuries occurred most commonly during military combat training and physical training, due to falls, soft tissue injuries with no object being handled.

## Figures and Tables

**Table 1 ijerph-16-00012-t001:** Body sites of SPIs and MPIs reported by ARES personnel. Reported as number of injuries (percentage of overall injuries).

Location	MPI	SPI	MPI and SPI
Knee	224 (16.4%)	4 (6.1%)	228 (15.9%)
Ankle	139 (10.2%)	2 (3.0%)	141 (9.8%)
Foot	55 (4.0%)	2 (3.0%)	57 (4.0%)
Thigh	46 (3.4%)	0	46 (3.2%)
Lower limb general	35 (2.6%)	2 (3.0%)	37 (2.6%)
Gastroc-soleus complex	25 (1.8%)	0	25 (1.7%)
Hip	18 (1.3%)	1 (1.5%)	19 (1.3%)
Shins	15 (1.1%)	0	15 (1.1%)
Lower limb multiple	13 (1.1%)	0	13 (0.9%)
Pelvis	1 (0.1%)	0	1 (0.1%)

Data expressed as number of injuries (% of injuries). MPI = minor personal injury, SPI = serious personal injury.

**Table 2 ijerph-16-00012-t002:** Nature of injuries of knee injuries in Australian Army Reserve (ARES) personnel.

Nature of Injury	MPI	SPI
Soft tissue injuries (trauma or unknown mechanisms)	181 (80.8%)	0
Trauma to joints and ligaments	16 (7.1%)	0
Contusion/bruising and superficial crushing	8 (3.6%)	0
Dislocation	6 (2.7%)	2 (50%)
Laceration	5 (2.2%)	0
Superficial injury	5 (2.2%)	0
Infectious and parasitic diseases	1 (0.4%)	0
Fracture	0	0
Multiple injuries	1 (0.4%)	0
Other diseases of skin and subcutaneous tissue	1 (0.4%)	0
Bursitis	0	1 (25%)
Other soft tissue diseases	0	1 (25%)
Total	224	4

Data expressed as number of injuries (% of injuries). MPI = minor personal injury, SPI = serious personal injury.

**Table 3 ijerph-16-00012-t003:** The activity in which soft tissue injuries occurred at the knee in part-time personnel.

Activity	MPI
Combat training	77 (42.2%)
Physical training	53 (29.5%)
Marching	14 (8.7%)
Walking	13 (6.9%)
Boarding/alighting a vehicle	5 (2.9%)
Manual/materials handling	4 (2.3%)
Sitting/standing	3 (1.7%)
Admin activities	2 (1.2%)
Driving	2 (1.2%)
Sport	3 (1.7%)
Construction/engineering	1 (0.6%)
Diving/immersion training	1 (0.6%)
Maintenance	1 (0.6%)
Sleeping	1 (0.6%)
Adventure training	1 (0.6%)
Total	181

Data expressed as number of injuries (% of injuries). MPI = minor personal injury.

**Table 4 ijerph-16-00012-t004:** Knee injuries suffered during combat training. Number of injuries (% of injuries).

Mechanism	MPI
Falls	28 (36.4%)
Muscular stress with no object being handled	24 (31.2%)
Muscular stress while lifting carrying or putting	5 (6.5%)
Contact with moving or stationary object	13 (16.9%)
Other and multiple mechanisms	5 (6.5%)
Contact with or exposure to biological factors of non-human origin	1 (1.3%)
Stepping kneeling or sitting on objects	1 (1.3%)
Total	77

Data expressed as number of injuries (% of injuries). MPI = minor personal injury.

**Table 5 ijerph-16-00012-t005:** Knee injuries suffered during physical training. Number of injuries (% of injuries).

Mechanism	MPI
Muscular stress with no object being handled	38 (71.7%)
Falls	8 (15.1%)
Muscular stress while lifting carrying or putting	1 (1.9%)
Contact with moving or stationary object	4 (7.5%)
Other and multiple mechanisms	2 (3.8%)
Total	53

Data expressed as number of injuries (% of injuries). MPI = minor personal injury.

**Table 6 ijerph-16-00012-t006:** Ages of ARES personnel who reported knee injuries (*n* = number of injuries).

Age (years)	*n*	%
<20	7	3.1%
20–29	99	43.4%
30–39	45	19.7%
40–49	53	23.3%
≥50	24	10.5%
Total	228	100.0%

**Table 7 ijerph-16-00012-t007:** Military ranks of ARES personnel who reported knee injuries (*n* = number of injuries).

Rank	*n*	%
Recruit	25	11.0%
Officer cadet	26	11.4%
Private trainee	6	2.6%
Private	44	19.3%
Private proficient	51	22.4%
Lance corporal	10	4.4%
Corporal	26	11.4%
Sergeant	11	4.8%
Warrant officer class 2	7	3.1%
Warrant officer class 1	3	1.3%
Lieutenant	4	1.8%
Captain	7	3.1%
Major	5	2.2%
Lieutenant colonel	2	0.9%
Colonel	1	0.4%
Total	228	100.0%

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
