# Peer review of "A Profile of Knee Injuries Suffered by Australian Army Reserve Soldiers"

_ijerph, 2018, doi:10.3390/ijerph16010012_

Round 1
Reviewer 1 Report
Overall Comments
This is generally a well-written and conducted study that provides injury information specifically for part-time Soldiers. This is a current gap in the literature.
Specific Comments
Abstract
Check for spelling and typos. Don’t list key words that are in your title
Introduction
There are some wording issues here. Try to keep writing straight forward and simple for the reader.
line 29, remove “therefore placed”
Line 34 – try removing “Similar to the military forces of many nations,” for a better first sentence
Line 37 – change “to those of” to “as”
Line 38, insert a ; after personnel
Line 39-41 – work on wording here for clarity
Line 44 – try removing “it has previously……to full time personnel,”
Line 46 – end sentence after [9]. start next sen at “It is wll known…” Also – ref 11 doesn’t seem appropriate
Line 50 – try removing “and especiall part time personnel”
Line 54 – Natures and causes should be singular
Line 59 – remove “on this basis,”
Line 60 – add Lower limb after the word common
62 – remove “,specifically for the lower limb”
Methods
line 66 – adding the N here would be helpful
line 72 – try moving “to which the injured solder belonged” and “injured soldiers’ “
Line 73 – change ‘the date on which the injury occurred” to “injury date”
Line 74 – remove “affected by the injury”
Line 75 – remove the word “incident’ after injury
Line 81 – end the sentence after [18]. then start with “SPI”
Line 89, remove the 2 commas
Line 91 - add a ; after ‘removed”
Line 95 – add a ; after “accordingly”
There is no statistical section
Results
Stats are simplistic but straight forward
Line 147-148 - this section is confusing. Are you just simply reporting other injuries in other papers? Is there enough data to break this up? If the reasoning is the same for all injuries they should really be in one bigger paper.
Line 152-153 – this sentence is passive, revise.
Line 164 – this is really a ‘catch all ‘ category. you should note this (several places) and explain that it is likely large for that reason
Discussion
Generally well written
line 224 – see previous comment about this catch-all category
Line 237 – end after “prepared for.” then start with “Unlike…”
Line 238 – remove comma after “training’
Lines 253-254 – is this really a concern? is this really the reason they don’t train?
Line 257 – consider changing ‘leading’ to ‘common’. Leading seems misleading….
Lines 267 – 269 – wording is confusing
Line 278-279 – remove this sentence – it is not needed.
Line 284 – relate this to knee injury
Line 300 – consider changing ‘it is known that…” to one study showed….. several of these in this section are one small study, careful what you assume
a table comparing full time to part time injuries might be a good addition to serve your point
Paragraph at 313 – this is confusing – please re-work this
Conclusion
This is written as a summary, not a conclusion. If you want a conclusion simply state what this study found, then put your summary statements at the end of the discussion.
Tables/Figures
Typically tables include a Note: under them with a listing of all abbreviations. Also, units for the # need to be described.
Why include injures when the total # is 0 or they don’t relate to the knee? (ie table 2, disc displacement isn’t a knee injury)
make sure all tables formatted the same and to journal requirements (ie bolding, font size, etc)
Author Response
Reviewer 1:
Overall Comments
This is generally a well-written and conducted study that provides injury information specifically for part-time Soldiers. This is a current gap in the literature.
Thank you for your comments regarding our manuscript. Please find all your specific comments addressed below.
Specific Comments
Abstract
Check for spelling and typos. Don’t list key words that are in your title
Thank you for your comments. This has been amended in the manuscript.
Introduction
There are some wording issues here. Try to keep writing straight forward and simple for the reader.
line 29, remove “therefore placed”
Line 34 – try removing “Similar to the military forces of many nations,” for a better first sentence
Line 37 – change “to those of” to “as”
Line 38, insert a ; after personnel
Thank you, these have all been amended in the manuscript.
Line 39-41 – work on wording here for clarity
Thank you for your comment, this has been changed to Injuries may be of even greater concern for part time soldiers, as previous research has shown part-time personnel report a substantially higher rate of injuries per unit exposure to active duty when compared to their full-time counterparts, with some showing up to double the rates in part-time compared to full-time personnel [1,2]
Line 44 – try removing “it has previously……to full time personnel,”
Thank you for your comment, this has been changed in the manuscript.
Line 46 – end sentence after [9]. start next sen at “It is wll known…” Also – ref 11 doesn’t seem appropriate
This has been changed in the manuscript, apologies for the reference which has been changed, right author, wrong paper!
Line 50 – try removing “and especiall part time personnel”
Line 54 – Natures and causes should be singular
Line 59 – remove “on this basis,”
Line 60 – add Lower limb after the word common
62 – remove “,specifically for the lower limb”
Thank you for your comments, these have all been amended in the manuscript.
Methods
line 66 – adding the N here would be helpful
line 72 – try moving “to which the injured solder belonged” and “injured soldiers’ “
Line 73 – change ‘the date on which the injury occurred” to “injury date”
Line 74 – remove “affected by the injury”
Line 75 – remove the word “incident’ after injury
Line 81 – end the sentence after [18]. then start with “SPI”
Line 89, remove the 2 commas
Line 91 - add a ; after ‘removed”
Line 95 – add a ; after “accordingly”
There is no statistical section
Thank you for your comments pertaining to the methods section, these have all been addressed in the manuscript.
Results
Stats are simplistic but straight forward
Line 147-148 - this section is confusing. Are you just simply reporting other injuries in other papers? Is there enough data to break this up? If the reasoning is the same for all injuries they should really be in one bigger paper.
The focus of this paper was the knee as it is the most commonly injured location in this population, however we did want to acknowledge other factors. To avoid confusion, this section has been revised.
Line 152-153 – this sentence is passive, revise.
Thank you for your comment, this has been revised.
Line 164 – this is really a ‘catch all ‘ category. you should note this (several places) and explain that it is likely large for that reason
Agreed, this has been addressed in the manuscript. However, the nature of injury being soft tissue was the most common and therefore important for injury reduction programs. The following has been added… It should be acknowledged however that the categorization of the nature of injury soft tissue injuries (trauma or unknown mechanisms) may be broad in nature and therefore may be over-represented because of this.
Discussion
Generally well written
line 224 – see previous comment about this catch-all category
See previous response.
Line 237 – end after “prepared for.” then start with “Unlike…”
Line 238 – remove comma after “training’
Thank you for your comments, these have been changed in the manuscript.
Lines 253-254 – is this really a concern? is this really the reason they don’t train?
To be blunt… yes. It is not the reason they don’t train; however, it does make it more difficult to obtain the specificity of training for combat training for part time personnel. Recently a soldier who went for a pack march while carrying no weapons or ammunition sent schools in to lock down and triggered a response from 15 patrol cars and a police helicopter due to fearful public reports. He was charged with public nuisance (charges later dropped). https://www.brisbanetimes.com.au/national/queensland/adf-apologises-after-soldiers-training-causes-alarm-20151202-glduz7.html
Line 257 – consider changing ‘leading’ to ‘common’. Leading seems misleading….
Thank you for your comment, this has been changed.
Lines 267 – 269 – wording is confusing
This has been changed to read ‘The most common mechanisms of injury during Combat Training and Physical Training were falls and muscular stress with no objects being handled.’
Line 278-279 – remove this sentence – it is not needed.
Thank you, it has been removed.
Line 284 – relate this to knee injury
The following sentence has been added Poor landing mechanics have been linked to acute ankle and knee sprains [3]
Line 300 – consider changing ‘it is known that…” to one study showed….. several of these in this section are one small study, careful what you assume
This has been changed to “There is some evidence suggesting…” to be less definitive.
a table comparing full time to part time injuries might be a good addition to serve your point
While we appreciate this suggestion, the injuries are quite literally exactly the same. We have published this previously [1,4,5]. Part time personnel tend to suffer more combat training injuries while full time tend to suffer more physical training injuries. Given that there are already seven tables in this manuscript, we have decided to not include this table.
Paragraph at 313 – this is confusing – please re-work this
This has been changed to read “A recent study has suggested that only 11-19% of injuries are captured with the WHSCAR system [1]. Under this system, the casualty is required to report to a system retrospectively and therefore the numbers of injuries reported in this study may underrepresent the absolute number of injuries in this population. It remains likely the proportional distributions of body sites, natures and causes of injuries reported by ARES personnel are still representative of this part-time army population. There have been recommendations for a ‘point of care’ system of reporting to be utilized, in which health care personnel create a report at the time an injured soldier presents for health care, rather than relying on the casualty to report to a system retrospectively [1].
Conclusion
This is written as a summary, not a conclusion. If you want a conclusion simply state what this study found, then put your summary statements at the end of the discussion.
The conclusion has been revised to state the studies findings.
Tables/Figures
Typically tables include a Note: under them with a listing of all abbreviations. Also, units for the # need to be described.
Thank you for your comment, this has been added.
Why include injures when the total # is 0 or they don’t relate to the knee? (ie table 2, disc displacement isn’t a knee injury)
Apologies for the oversight, these have been deleted.
make sure all tables formatted the same and to journal requirements (ie bolding, font size, etc)
Thank you, this has been changed in the manuscript.
1. Pope, R.; Orr, R. Incidence rates for work health and safety incidents and injuries in australian army reserve vs full time soldiers, and a comparison of reporting systems. Journal of Miltary and Veterans Health 2017, 25, 16-25.
2. Department of Defence. Australian defence force health status report. Department of defence. Http://www.Defence.Gov.Au/media/2000/health/01.Pdf. 2000.
3. McKay, G.D.; Goldie, P.A.; Payne, W.R.; Oakes, B.W. Ankle injuries in basketball: Injury rate and risk factors. British Journal of Sports Medicine 2001, 35, 103.
4. Schram, B.; Orr, R.; Pope, R. Injuries in australian army full-time and part-time personnel undertaking basic training JMVH 2018, (submitted).
5. Orr, R.M.; Pope, R.; Schram, B.; MacDonald, D.; Hing, W. Profiling work health and safety incidents and injuries in australian army personnel: An investigation of injuries and other incidents suffered by army reserve personnel; ePublications@bond: 2016.
Reviewer 2 Report
In this manuscript, the authors report on a study in order to define a profile of knee injuries suffered by Australian Army Reserve Soldiers.
Among 1434 of injuries reported by ARES personnel were analyzed in order to profile the leading body site of injury occurring in part time soldiers to inform injury prevention strategies. The authors report that most commonly soft tissue injuries affect the knee during military combat training and physical training, due to falls, soft tissue injuries with no object being handled.
The work is interesting and present relevant information to define injury prevention strategies.
In order to improve the manuscript, please considered the following comments.
1. As mentioned earlier, the text seems into the analysis about records of injuries reported by Australian Army Reserve (ARES) personnel over a two-year period, however the procedure do not consider gender. I think that is a crucial point and must be described. Also is not clear how the data was analyzed if the same person presents different injury cases during the two-year period.
2. In Table 1 in location column is indicated: “Multiple”, “Lower Limb general”, “Lower Limb multiple”, but is not presented a description of these locations. This must be clearer.
3. Review the percentages on lines 147 and 148. Describe better these percentages.
4. In line 159 describe injury results in the back, this information is presented in table 1?, please review.
5. In conclusions, line 326 is indicated the use of ankle bracing, why not also knee bracings?
Minor
1. In line 199must be Table 7
2. In table 2 a parenthesis is missing
3. anatomical sites = body sites? is necessary to standardize the sentences to avoid confuse at the reader.
Author Response
In this manuscript, the authors report on a study in order to define a profile of knee injuries suffered by Australian Army Reserve Soldiers.
Among 1434 of injuries reported by ARES personnel were analyzed in order to profile the leading body site of injury occurring in part time soldiers to inform injury prevention strategies. The authors report that most commonly soft tissue injuries affect the knee during military combat training and physical training, due to falls, soft tissue injuries with no object being handled.
The work is interesting and present relevant information to define injury prevention strategies.
In order to improve the manuscript, please considered the following comments.
1. As mentioned earlier, the text seems into the analysis about records of injuries reported by Australian Army Reserve (ARES) personnel over a two-year period, however the procedure do not consider gender. I think that is a crucial point and must be described. Also is not clear how the data was analyzed if the same person presents different injury cases during the two-year period.
Thank you for raising this valid point. If someone was to present with a different injury during the two year period, it would still be included in the analysis as it is still an injury suffered during military service. Gender was not reported in the database; therefore we are unable to comment on it.
2. In Table 1 in location column is indicated: “Multiple”, “Lower Limb general”, “Lower Limb multiple”, but is not presented a description of these locations. This must be clearer.
Good point. The location ‘multiple’ refers to multiple body locations including those above the lower limb, therefore it should not have been included in the table. It has now been removed.
3. Review the percentages on lines 147 and 148. Describe better these percentages.
In line with reviewer one’s comment, this has been revised.
4. In line 159 describe injury results in the back, this information is presented in table 1?, please review.
Thank you for your comment, this has been removed.
5. In conclusions, line 326 is indicated the use of ankle bracing, why not also knee bracings?
Thank you for your comment. We suggested ankle bracing as studies from the US have shown a benefit on reducing ankle injuries with braces. At this stage there is no evidence for knee braces to be effective in reducing military injury rates. In addition, most ankle injuries are traumatic in nature, highlighting the rationale for a brace, as opposed to the cumulative, soft tissue injuries at the knee.
Minor
1. In line 199must be Table 7
Thank you for pointing this out, this has been amended.
2. In table 2 a parenthesis is missing
Thank you, this has been added.
3. anatomical sites = body sites? is necessary to standardize the sentences to avoid confuse at the reader.
Thank you, this has been changed to body sites.
Reviewer 3 Report
Comments in the attached file.

Author Response
General
The current manuscript looks to determine the leading body site of injury and primary cause of
injury in part-time soldiers within the Australian Army Reserve. The specific purpose within this
manuscript was to identify leading injuries in the lower extremity. The research fills a gap in the
current literature on army reservist in relation to injury rates and causes. While the manuscript is
well written, I do believe it needs significant revisions before it can be published.
COMMENTS
General
Throughout the abstract and manuscript, you alternate between hyphenating full-time and part time and not hyphenating. Please be consistent and hyphenate when grammatically correct.
Thank you for your comment, this has been addressed in the manuscript.
Abstract
Page 1, Line 18: Sentence doesn’t make sense and should end with a period.
Thank you, this has been changed in the manuscript.
Page 1, Line 21: Target, intrinsic and extrinsic approached… I believe you mean approaches.
Thank you, this has been changed in the manuscript.
Introduction
Page 1, Line 38-41: This sentence is confusing and long. Consider rewriting and dividing into 2
sentences.
In line with reviewer one’s comment, this has been changed in the manuscript.
Page 1, Line 44-47: This sentence is very long and doesn’t read well. Split into 2 different
sentences.
In line with reviewer one’s comment, this has been changed in the manuscript.
Methods
Page 3 Line 107-114: Your explanation is extremely confusing about your calculations for injury
incidence. You also never make any rational for why you do this calculation for full-time
soldiers when examining part-time reservist.
The calculation relative to injury incidence was made to compare to full-time injury rates. As the part-time personnel have less exposure than full-time, absolute comparisons would give a false representation of the figures. The following comparison has been added to further this rationale. Part-time personnel (30.5 injuries per 100 years of active service) appear to be injured more than their full-time colleagues with previous reports of full-time personnel injury rates of 16.72 injuries per 100 years of active service [1]).
Page 3, Line 133 & 134: Authorisation should be authorization.
Thank you for your comment, his has been changed in the manuscript.
Results
Page 4, Line 147-148: Not sure where you came up with the 38.4% of MPI and 37.2% of all
injuries. That doesn’t match anything in Table 1. You may need to explain this further for the
reader to make a connection if those numbers are correct.
In line with reviewer one’s comments, this has been changed.
Page 4, Line 152-153: “Due to the knee being the most…”. In the next paragraph you talk about
the second most common area of injury is the back. The back isn’t listed anywhere in your tables
and most do not include back injuries as part of the lower extremity. Just remove that
information.
This has been removed.
Page 4, Table 2: You are missing a close parenthesis ) after the 80.8%.
Thank you, this has been added.
Page 6, Line 184-195: Your discussion on rank is confusing. Either report this data in relation to
years of service, or combine the data and report as trainee, enlisted, or officer. This probably
makes the most sense since these 3 categories have significantly different levels of fitness
requirements to complete their jobs.
Thank you for your comment. This has been changed to compare officers and non-officers. In reserve personnel, the recruits and officer cadets complete the same basic training, therefore have the same fitness requirements and assessments.
Discussion
Page 7 Line 207: While your % match Table 1, they do not match the information mentioned in
the comment above. This leads to confusion during reading.
Apologies for the confusion, the paragraph will the incorrect percentages has been removed.
Page 7, Line 215: “…findings of this study in which the ankle was…” Just missing the ‘t’ in the.
Thank you this has been added.
Page 7, Line 225: remove additional space between military personnel, most commonly involve.
Thank you this has been removed.
Page 7, Line 226: add a comma between ‘meniscus’ and ‘and ligaments’.
Thank you this has been added.
Page 7, Line 228: add a comma between ‘infantry service’ and ‘and prior hip injury’.
Thank you this has been added.
Page 8, Lines 235-254: You use the word “carriage” several times. While the use is technically
correct, the word “carriage” is not the common English language word that would be used in
these cases. Recommend revising sentences to use the word ‘carry’ or ‘carrying’.
Thank you for your comment however, respectfully, we disagree. In the military context, the term is load carriage. It has been used extensively throughout the literature in Australia, the US and UK [2-17].
Page 8, Line 239: What is a preferably tactical scenarios? Consider revising sentence to make
this more understandable.
Thank you for this comment, this has been changed to read military styled tasks.
Page 8, Line 239: Delete the extra space between sentences.
Thank you for pointing this out, it has been removed.
Page 8, Line 261: Remove the word ‘chronic’ from the sentence.
Thank you this has been removed.
1. Pope, R.; Orr, R. Incidence rates for work health and safety incidents and injuries in australian army reserve vs full time soldiers, and a comparison of reporting systems. JMVH 2017, 25, 16-25.
2. Orr, R. Soldier load carriage: A risk management approach. The University of Queensland, Australia, 2013.
3. Orr, R.; Johnston, V.; Coyle, J.; Pope, R. Reported load carriage injuries of the australian army soldier. J Occup Rehabil. 2015, 25, 316-322.
4. Orr, R.; Pope, R.; Johnston, V.; Coyle, J. Soldier self-reported reductions in task performance associated with operational load carriage. JASC 2013, 21, 39-46.
5. Orr, R.M.; Coyle, J.; Johnston, V.; Pope, R. Self-reported load carriage injuries of military soldiers. International Journal of Injury Control & Safety Promotion 2017, 24, 189-197.
6. Orr, R.M.; Johnston, V.; Coyle, J.; Pope, R. Reported load carriage injuries of the australian army soldier. J Occup Rehabil 2015, 25, 316-322.
7. Orr, R.M.; Pope, R. Gender differences in load carriage injuries of australian army soldiers. BMC Musculoskeletal Disorders 2016, 17, 1-8.
8. Orr, R.M.; Pope, R.; Johnston, V.; Coyle, J. Soldier occupational load carriage: A narrative review of associated injuries. Int J Inj Contr Saf Promot 2013, 1-9.
9. Orr, R.M.; Pope, R.R. Load carriage: An integrated risk management approach. J Strength Cond Res 2015, 29 Suppl 11, S119.
10. Knapik, J.; Reynolds, K. Load carriage in military operations: A review of historical, physiological, biomechanical, and medical aspects. 1997.
11. Knapik, J.J.; Harman, E.A.; Steelman, R.A.; Graham, B.S. A systematic review of the effects of physical training on load carriage performance. J Strength Cond Res 2012, 26, 585-597.
12. Knapik, J.J.; Reynolds, K.L.; Harman, E. Soldier load carriage: Historical, physiological, biomechanical, and medical aspects. Mil Med 2004, 169, 45.
13. Blacker, S.D.; Fallowfield, J.L.; Bilzon, J.L.; Willems, M.E. Neuromuscular impairment following backpack load carriage. J Hum Kinet 2013, 37, 91-98.
14. Blacker, S.D.; Fallowfield, J.L.; Bilzon, J.L.; Willems, M.E. Neuromuscular function following prolonged load carriage on level and downhill gradients. Aviat Space Environ Med 2010, 81, 745-753.
15. Blacker, S.D.; Williams, N.C.; Fallowfield, J.L.; Willems, M.E. The effect of a carbohydrate beverage on the physiological responses during prolonged load carriage. Eur J Appl Physiol 2011, 111, 1901-1908.
16. Fallowfield, J.L.; Blacker, S.D.; Willems, M.E.; Davey, T.; Layden, J. Neuromuscular and cardiovascular responses of royal marine recruits to load carriage in the field. Appl Ergon 2012, 43, 1131-1137.
17. Billing, D.C.; Silk, A.J.; Tofari, P.J.; Hunt, A.P. Effects of military load carriage on susceptibility to enemy fire during tactical combat movements. J Strength Cond Res 2015, 29 Suppl 11, S134-138.